# LAPO: Internalizing Reasoning Efficiency via Length-Adaptive Policy Optimization

## Abstract

Large reasoning models have achieved remarkable performance through extended chain-of-thought sequences, yet this computational freedom leads to excessive token generation even for simple problems. We present Length-Adaptive Policy Optimization (LAPO), a novel framework that transforms reasoning length control from an external constraint into an intrinsic model capability. Unlike existing approaches that impose rigid limits or rely on post-hoc interventions, LAPO enables models to internalize an understanding of appropriate reasoning depth through a two-stage reinforcement learning process. In the first stage, models learn natural reasoning patterns by discovering the statistical distribution of successful solution lengths. In the second stage, these learned patterns are embedded as in-context, self-declarative guidance, teaching the model to proactively plan its reasoning budget. Experiments on mathematical reasoning benchmarks demonstrate that LAPO reduces token usage by up to 40.9% while improving accuracy by 2.3%. Our analysis reveals that models trained with LAPO develop emergent abilities to allocate computational resources based on problem complexity, achieving efficient reasoning without sacrificing quality.

## 1 Introduction

Recent advances in large reasoning models have demonstrated remarkable capabilities through extended chain-of-thought sequences Wei et al. (2022); Jaech et al. (2024); DeepSeek-AI et al. (2025). However, this computational freedom leads to "overthinking" Min et al. (2024): models generate excessively verbose reasoning chains even for simple problems, causing significant computational overhead and hindering practical deployment.

Existing approaches to address this challenge fall into three main categories, each with inherent limitations. Direct length reduction methods either rely on reward design Yang et al. (2025); Huang et al. (2025) that can cause over-shortening and accuracy degradation, or impose hard length constraints Aggarwal & Welleck (2025); Hou et al. (2025) that lack adaptability across problem types. Dynamic early-stopping approaches Qiao et al. (2025); Muennighoff et al. (2025) make real-time termination decisions but often truncate mid-reasoning, disrupting the thinking process. Adaptive thinking methods Lou et al. (2025); Zhang et al. (2025); Fang et al. (2025) enable models to switch between thinking and non-thinking modes but operate at a coarse granularity.

The fundamental limitation of these approaches is their treatment of length control as an external constraint. This very paradigm conflicts with the nature of mathematical reasoning, where intrinsic problem complexity alone should dictate the required reasoning depth. Current methods fail to recognize that when models successfully solve problems, they naturally converge to reasoning lengths reflecting this complexity. The challenge, therefore, is not to impose arbitrary limits, but to help models discover and internalize these natural patterns.

We propose a paradigm shift: instead of constraining reasoning through external mechanisms, we enable models to learn from their own successful reasoning patterns and develop an internal sense of appropriate reasoning depth. Our key insight is that the distribution of reasoning lengths in correct solutions contains valuable information about how much thinking each problem genuinely requires. By capturing these patterns during training and teaching models to anticipate the appropriate reasoning budget before they begin solving, we can transform length control from an external limitation into an intrinsic capability.

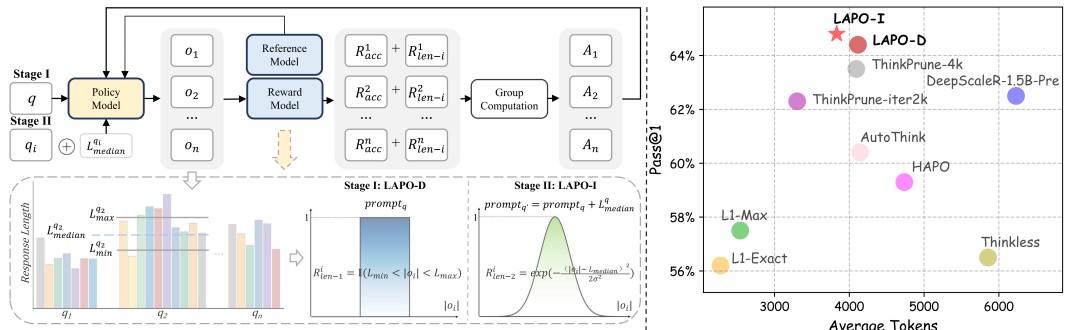

Figure 1: Overview of Length-Adaptive Policy Optimization (LAPO) and its superior performance. The LAPO framework (left) trains a model in two stages: first discovering natural reasoning lengths, then internalizing them as self-proposed budgets. This process enables our models (LAPO-I) to achieve a state-of-the-art balance between accuracy and efficiency (right), surpassing existing methods by operating in the desirable top-left region of the performance plot.

We introduce Length-Adaptive Policy Optimization (LAPO), a two-stage reinforcement learning framework that transforms length control into an intrinsic model capability. As illustrated in Figure 1, LAPO first operates in a Discovery stage, where a length-aware reward encourages the model to find a robust distribution of efficient yet correct solution lengths. This moves beyond simply rewarding the shortest answer by identifying a zone of reasonableness. The pivotal second stage, Internalization, embeds these discovered statistical patterns (specifically, the median length) as in-context, self-declarative guidance (e.g., ...`<think>` I will answer with n tokens). This technique reframes the budget not as an external command, but as part of the model's own reasoning plan, teaching it to proactively allocate its computational resources.

LAPO fundamentally differs from existing approaches by recognizing that true efficiency stems from understanding problem-specific computational needs, not from following rigid rules. Our two-stage design enables a natural progression: models first learn appropriate reasoning depth through experience, then internalize this knowledge to proactively anticipate task demands. This process mirrors how human experts develop intuition, allocating mental effort in proportion to a problem's complexity.

Extensive experiments validate the effectiveness of our approach. LAPO achieves remarkable efficiency gains, reducing token usage by up to 40.9% while simultaneously improving accuracy by 2.3% on mathematical reasoning benchmarks (see Figure 1). Our analysis reveals that this improvement stems from the model's ability to distinguish between problems requiring elaborate derivations versus those needing only brief calculations. These results indicate that when models learn from their own successful patterns rather than arbitrary constraints, they develop more robust and efficient reasoning strategies.

Our main contributions are:

- We propose LAPO, a novel two-stage RL framework that transforms length control from an external constraint into an intrinsic, adaptive capability by learning from the model's own successful reasoning.

- We introduce a training method that uses discovered statistical patterns as in-context, self-declarative guidance, enabling models to internalize efficient reasoning behaviors without sacrificing inference-time flexibility.

- We demonstrate that LAPO achieves substantial efficiency gains (up to 40.9% token reduction) while simultaneously improving accuracy, revealing a robust capability for adaptive resource allocation.

## 2 RELATED WORKS

### 2.1 TEST-TIME SCALING IN LARGE LANGUAGE MODELS

Increasing test-time computation has consistently been shown to improve performance in complex reasoning tasks, mathematical problem-solving, and code generation Wu et al. (2025); Wang et al. (2023); Wei et al. (2022); DeepSeek-AI et al. (2025). Test-time scaling laws indicate predictable performance gains from increasing inference computation, either by generating more reasoning chains or longer ones Wu et al. (2025); Snell et al. (2024); Jaech et al. (2024). Prominent approaches include parallel sampling of multiple reasoning paths Wang et al. (2023), tree-based search Yao et al. (2023); Wu et al. (2025), and iterative refinement techniques Snell et al. (2024); Welleck et al. (2024).

Recent reasoning models such as OpenAI's O1 and DeepSeek's R1-style models Jaech et al. (2024); DeepSeek-AI et al. (2025) simplify test-time scaling by generating extended reasoning traces through reinforcement learning with verifiable rewards (RLVR), encouraging deep thinking behaviors such as broad exploration and feasibility checks Gandhi et al. (2025). However, these extended reasoning behaviors often lead to much longer reasoning traces, sometimes several times longer than those produced by short CoT models Sui et al. (2025); Chen et al. (2024), creating an "overthinking" issue that largely increases inference costs Kumar et al. (2025).

### 2.2 EFFICIENT LONG CHAIN-OF-THOUGHT LLM

To address overthinking, various methods have been proposed. Prompt-based methods offer imprecise control Xu et al. (2025a). Training-based methods, using supervised fine-tuning Wang et al. (2024); Kang et al. (2025); Ma et al. (2025); Xia et al. (2025) or RL with length penalties Muennighoff et al. (2025); Chang et al. (2025); Luo et al. (2025); Xu et al. (2025b), often fail to adapt to problem complexity. Router-based methods add computational overhead by routing queries between models Chuang et al. (2025); Ong et al. (2024). While recent approaches like L1 Aggarwal & Welleck (2025) and Elastic Reasoning Xu et al. (2025b) can adhere to a given token budget, they cannot autonomously estimate an appropriate budget for a given problem.

In contrast, our LAPO framework is designed to address this gap. Through its two-stage "Discover-Internalize" process, LAPO explores a new direction where models learn to perform both autonomous budget estimation and problem-adaptive length control. By training models to learn from their own successful reasoning patterns, our approach aims to bridge the gap between high-quality reasoning and computational efficiency in a way that prior work has not.

## 3 METHOD

We present Length-Adaptive Policy Optimization (LAPO), a framework designed to transform efficient reasoning from an externally imposed constraint into an intrinsic model capability. Our approach is built on the insight that the distribution of lengths across successful solutions reflects a problem's intrinsic complexity. LAPO leverages these patterns in a two-stage process: it first Discovers natural reasoning lengths via a length-aware reward, then Internalizes this knowledge by training the model to follow its own self-declarative reasoning plan, as illustrated in Figure 2.

### 3.1 DISCOVERY STAGE: LEARNING NATURAL REASONING PATTERNS

The Discovery stage aims to uncover inherent relationships between problems and their natural reasoning lengths through GRPO training with a carefully designed reward mechanism that encourages efficient exploration while maintaining correctness.

**Extracting Statistics from GRPO Rollouts.** During GRPO training, we generate $N$ rollout responses for each problem $q$ in the training batch. From these rollouts, we collect the lengths of responses that produce correct answers:

$$\mathcal{L}_q = \{|r_i| : \mathbb{I}(y_i = y_{\text{gold}}) = 1, i \in [1, N]\} \tag{1}$$

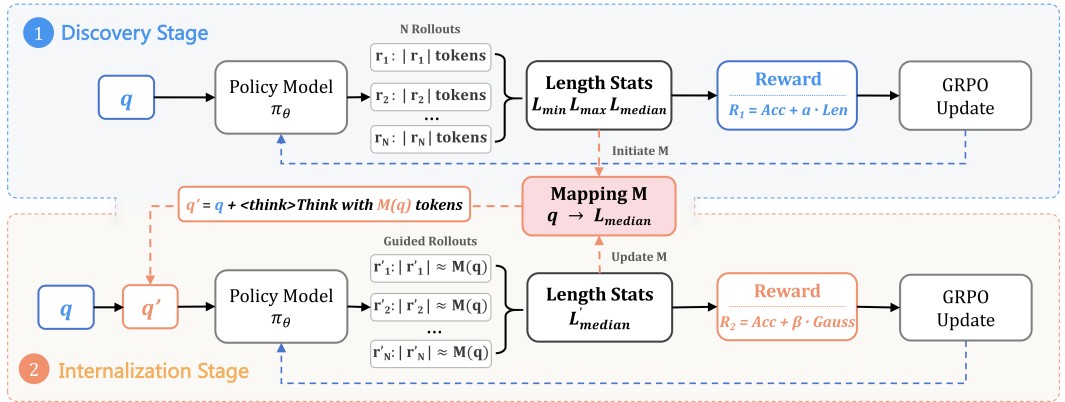

Figure 2: The LAPO framework consists of two stages: (1) Discovery stage learns natural reasoning patterns by rewarding efficient correct solutions and collecting length statistics; (2) Internalization stage embeds these statistics as self-proposed plans within the model's reasoning context, teaching models to internalize efficient reasoning.

where $y_i$ is the predicted answer from the $i$-th rollout response $r_i$. This collection, extracted directly from the GRPO sampling process, represents natural variation in successful reasoning lengths.

We derive two key statistics from these rollouts. First, we establish a reasonable length range using percentiles to filter outliers while preserving central tendencies:

$$[L_{\min}, L_{\max}] = [\text{Percentile}_{30}(\mathcal{L}_q), \text{Percentile}_{70}(\mathcal{L}_q)] \tag{2}$$

This choice is designed to robustly identify the core distribution of effective reasoning lengths by filtering out statistical outliers. The lower bound (30th percentile) helps discard overly concise solutions that might be correct by chance, while the upper bound (70th percentile) discourages excessively verbose and inefficient reasoning paths.

Second, we create a problem-to-length mapping that will guide the Internalization stage:

$$\mathcal{M} : q \mapsto L_{\text{median}}(q) = \text{Median}(\mathcal{L}_q) \tag{3}$$

For problems without correct solutions in the current rollouts, we temporarily set $\mathcal{M}(q) = 4096$ (maximum sequence length) to encourage comprehensive exploration in subsequent episodes. This high initial budget is a fallback measure that is promptly updated to the data-driven median once the model starts solving the problem (Eq. 8), preventing a lasting bias toward long-form answers.

**Length-Aware Reward Design.** We employ a composite reward function balancing accuracy and efficiency:

$$R_D(r_i, q) = \mathbb{I}(y_i = y_{\text{gold}}) + \alpha \cdot R_1(r_i, q) \tag{4}$$

The length component operates on a crucial principle—only correct responses receive length-based rewards. Let $\mathcal{C}_i = \mathbb{I}(y_i = y_{\text{gold}})$ indicate whether the response is correct, and define the distance to the target length range as $d_i = \min(||r_i| - L_{\min}|, ||r_i| - L_{\max}|)$. We introduce a linear decay function $f(d) = \max(0, 1 - d/100)$ to penalize deviations from the efficient length range. The length reward is then defined as:

$$R_1(r_i, q) = \begin{cases} 1.0 & \text{if } \mathcal{C}_i = 1 \wedge |r_i| \in [L_{\min}, L_{\max}] \\ f(d_i) & \text{if } \mathcal{C}_i = 1 \wedge |r_i| \notin [L_{\min}, L_{\max}] \\ 0 & \text{if } \mathcal{C}_i = 0 \end{cases} \tag{5}$$

This design creates gradients guiding models toward efficient lengths while allowing flexibility for complex problems. Throughout the Discovery stage, we continuously update $\mathcal{M}$ after each GRPO training step to reflect evolving model capabilities.

---

**Algorithm 1** Length-Adaptive Policy Optimization(LAPO)

---

1: **Input:** Base model $\pi_\theta$, training data $\mathcal{D}$, hyperparameters $\alpha, \beta, \sigma, E_1, E_2$
2: **Output:** Length-adaptive model $\pi_\theta^*$
3:
4: **// Discovery Stage**
5: **for** episode $e = 1$ to $E_1$ **do**
6:     Sample batch $\mathcal{B} \subset \mathcal{D}$
7:     **for** each problem $q \in \mathcal{B}$ **do**
8:         Generate $N$ rollouts: $\{r_1, \ldots, r_N\} \sim \pi_\theta(q)$
9:         Collect correct lengths: $\mathcal{L}_q = \{|r_i| : y_i = y_{\text{gold}}\}$
10:        Compute range: $[L_{\min}, L_{\max}] = [\text{P}_{30}(\mathcal{L}_q), \text{P}_{70}(\mathcal{L}_q)]$
11:        Update mapping: $\mathcal{M}(q) = \text{Median}(\mathcal{L}_q)$
12:        Compute rewards: $R_D(r_i, q) = \mathbb{I}(y_i = y_{\text{gold}}) + \alpha \cdot R_1(r_i, q)$
13:     **end for**
14:     Update $\pi_\theta$ using GRPO with rewards $R_1$
15: **end for**
16:
17: **// Internalization Stage**
18: **for** episode $e = 1$ to $E_2$ **do**
19:     Sample batch $\mathcal{B} \subset \mathcal{D}$
20:     **for** each problem $q \in \mathcal{B}$ **do**
21:         Augment prompt: $q' \leftarrow q +$ "`<think>` I will answer the question with $\mathcal{M}(q)$ tokens."
22:         Generate $N$ rollouts: $\{r_1, \ldots, r_N\} \sim \pi_\theta(q')$
23:         Compute rewards: $R_I(r_i, q') = \mathbb{I}(y_i = y_{\text{gold}}) + \beta \cdot R_2(r_i, q')$
24:         Update mapping $\mathcal{M}(q)$ using dual-strategy (Eq. 8)
25:     **end for**
26:     Update $\pi_\theta$ using GRPO with rewards $R_2$
27: **end for**
28: **return** $\pi_\theta^*$

---

## 3.2 Internalization Stage: Length-Aware Efficient Reasoning

The Internalization stage transforms discovered patterns into internalized capabilities through continued GRPO training with modified prompts and rewards.

**Length-Conditioned Rollout.** We augment each problem prompt with explicit length guidance:

$$\text{prompt}'_q = \text{prompt}_q + \text{``<think> I will answer the question with } n \text{ tokens.''}$$

where $n = \mathcal{M}(q)$ from the Discovery stage. This embeds length awareness within the reasoning context, helping models perceive computational budgets as intrinsic to thinking rather than external constraints.

**Length-Adherence Reward.** To encourage the model to follow its self-declared reasoning budget, the Internalization stage employs a precision-focused reward function. This function is designed to reward the alignment between the model's output length and its self-declared budget n. The total reward is defined as:

$$R_I(r_i, q') = \mathbb{I}(y_i = y_{\text{gold}}) + \beta \cdot R_2(r_i, q') \tag{6}$$

where the adherence component, $R_2$, is only granted for correct solutions:

$$R_2(r_i, n) = \begin{cases} \exp\left(-\frac{(|r_i|-n)^2}{2\sigma^2}\right) & \text{if } \mathcal{C}_i = 1, \\ 0 & \text{if } \mathcal{C}_i = 0; \end{cases} \tag{7}$$

This Gaussian-inspired reward reinforces solutions that are both correct and consistent with the intended reasoning depth. The standard deviation $\sigma$ serves as a tolerance parameter, where a smaller $\sigma$ enforces stricter adherence and a larger $\sigma$ allows more flexibility. By rewarding adherence to the self-proposed plan, this mechanism guides the model to internalize the relationship between problem complexity and an appropriate computational budget, rather than merely tracking an external signal.

**Internalization via In-Context Guidance.** A cornerstone of our framework is how it fosters genuine internalization, enabling inference-time flexibility without explicit length targets. The key lies in the design of the augmented prompt. Placing the self-declarative guidance immediately after the `<think>` token transforms an external constraint into an intrinsic part of the model's cognitive plan.

During the Internalization stage, we refine $\mathcal{M}$ based on new GRPO rollouts with a dual-strategy update:

$$\mathcal{M}(q) \leftarrow \begin{cases} \text{Median}(\mathcal{L}_q^{(t)}) & \text{if unsolved} \\ \min(\mathcal{M}(q), \text{Median}(\mathcal{L}_q^{(t)})) & \text{if solved} \end{cases} \tag{8}$$

This ensures newly solved problems establish reasonable benchmarks while previously solved problems gravitate toward more efficient solutions.

### 3.3 TRAINING PIPELINE

We present the complete LAPO training procedure in Algorithm 1. LAPO employs GRPO across both stages with the following pipeline:

**Discovery Stage** (Lines 4-15): The model first learns natural reasoning patterns via GRPO with our length-aware reward. During this stage, we continuously update a problem-to-length mapping, $\mathcal{M}$, based on the statistics of successful rollouts, allowing the model to empirically discover problem-specific length distributions.

**Internalization Stage** (Lines 17-27): The model then learns to internalize these discovered patterns. We augment each prompt with the target length from $\mathcal{M}$ as in-context, self-declarative guidance inside the `<think>` block. An adherence-focused reward encourages the model to treat this budget as its own reasoning plan, while a dual-strategy update to $\mathcal{M}$ promotes continuous efficiency gains.

This progressive design mirrors cognitive development: first gaining tacit experience about appropriate reasoning depth through practice, then learning to anticipate these requirements proactively. The embedding of guidance as a self-declared plan is the very key mechanism that bridges this gap from experience to proactive anticipation, creating models that can intrinsically adapt computational effort to problem demands.

## 4 EXPERIMENT SETUP

**Training Details.** We train our models on a mixed dataset of 10,000 mathematical problems to ensure a balanced difficulty distribution, comprising 6,000 examples from the DeepScaleR-Preview-Dataset and 4,000 from the intermediate levels of the MATH dataset Hendrycks et al. (2021). We apply LAPO to two base models: DeepSeek-R1-1.5B DeepSeek-AI et al. (2025) and DeepScaleR-1.5B-Preview.

We train all models using the GRPO algorithm. Each of LAPO's two stages is trained for 3 episodes, with reward weights set to $\alpha$=0.7 and $\beta$=0.7 respectively. These values were chosen to provide a substantial efficiency signal without overpowering the primary reward for correctness. Training is conducted with a maximum context length of 4,096 tokens, a constraint also applied to relevant baselines like ThinkPrune and L1 to ensure a fair comparison. A comprehensive list of all hyperparameters is available in the Appendix. Note that we did not conduct extensive hyperparameter tuning, so one can expect further improvements with additional optimization.

**Evaluation Details.** At inference, we expand the generation window to a generous 32,768 tokens for all models to assess their true, unconstrained reasoning capabilities. This setup allows us to isolate the efficiency gains stemming directly from the LAPO framework, rather than from simple context window limitations. We evaluate on four challenging benchmarks: MATH-500 Hendrycks et al. (2021), AIME2024, AMC23, and Olympiad-Bench He et al. (2024). Following standard practices DeepSeek-AI et al. (2025), we report both Pass@1 accuracy and the average number

of tokens. For each problem, we sample N responses (4 for MATH-500/OlympiadBench, 32 for AIME/AMC) with a temperature of 0.6 and a top-p of 0.95.

**Baselines.** We benchmark LAPO against three classes of baselines: the foundational models, an ablation baseline, and existing methods designed for efficient reasoning. First, we evaluate the Base Models to establish a performance starting point. Second, to isolate the effect of our length-reward, we also include an Ablation Baseline, denoted as Acc-Only, which is trained with GRPO using only the accuracy reward. Finally, we compare against several state-of-the-art Efficient Reasoning Baselines, which represent different philosophies for achieving efficiency. (1)Implicit Regularization: HAPO Huang et al. (2025), which uses history-aware rewards. (2)Budget-Driven Control: L1 Aggarwal & Welleck (2025) and ThinkPrune Hou et al. (2025), which follow external length targets. (3)Adaptive Activation: AutoThink Tu et al. (2025), AdaptThink Zhang et al. (2025), and Thinkless Fang et al. (2025), which learn a binary think/no-think policy.

## 5 RESULTS AND ANALYSIS

We present comprehensive experimental results to validate LAPO's effectiveness and understand its underlying mechanisms. We first benchmark LAPO against state-of-the-art baselines (Section 5.1). We then conduct in-depth ablation studies on key design choices, including the the form of length guidance (Section 5.2) and the statistical metrics for target length selection (Section 5.3). Finally, we provide a qualitative analysis of the learned reasoning patterns (Section 5.4).

### 5.1 MAIN RESULTS

As shown in Table 1, LAPO achieves a superior balance of reasoning accuracy and computational efficiency, consistently outperforming its base models and establishing a new state-of-the-art frontier among methods that do not rely on external length controls.

**LAPO simultaneously enhances reasoning performance and reduces test-time computes.** Compared to its base models, LAPO delivers substantial gains. On DeepScaleR-1.5B-Preview, it reduces tokens by 38.5% while boosting average accuracy by 2.3 points; a similar trend holds for DeepSeek-R1-1.5B (41.0% token cut and 1.2 point accuracy gain). This validates that LAPO learns to produce more concise yet effective reasoning.

**LAPO surpasses existing efficient reasoning optimization approaches.** LAPO's effectiveness is further contextualized by comparison with existing paradigms. First, in contrast to budget-driven methods like ThinkPrune-4k, LAPO achieves higher accuracy under identical training conditions without needing an external length target at inference. Second, the comparison with implicit regularization methods like HAPO, which rewards the shortest correct solution, is particularly informative. Our results indicate that HAPO's token reduction is accompanied by a degradation in accuracy. LAPO, by targeting the median length, maintains or enhances performance, lending empirical support to the hypothesis that a statistically typical reasoning length is a more effective optimization target than the absolute minimum. Finally, while adaptive activation strategies like AutoThink are token-efficient, they do not attain LAPO's level of accuracy, suggesting that modulating reasoning length is a more effective mechanism for preserving performance than a binary think/no-think decision.

**Both Discovery and Internalization stages contribute to the final performance.** The framework's two-stage design is critical to these results. The Discovery stage (LAPO-D) establishes a strong initial policy, outperforming the accuracy-only baseline on both metrics and indicating the efficacy of its length-aware reward. The subsequent Internalization stage (LAPO-I) further refines this policy, using in-context guidance to cultivate a more deeply embedded adaptive reasoning capability.

### 5.2 ABLATION STUDY ON IN-CONTEXT GUIDANCE

To validate that our method's success stems from internalizing a self-proposed plan, we ablate the two key factors of our in-context guidance: its form (how precise the guidance is) and its position

Table 1: Main results on MATH500, AIME2024, AMC23, and OlympiadBench. We report Pass@1 accuracy (%) and the average number of generated tokens (#Tok). For each metric, **bold** indicates the best and underline indicates the second-best Pass@1 score within each base model group.

| | MATH-500 | | AIME2024 | | AMC-23 | | OlympiadBench | | Average | |
|---|---|---|---|---|---|---|---|---|---|---|
| | Pass@1 | #Tok | Pass@1 | #Tok | Pass@1 | #Tok | Pass@1 | #Tok | Pass@1 | #Tok |
| *Base model: DeepSeek-R1-1.5B* | | | | | | | | | | |
| HAPO | 82.2 | 2288 | 31.3 | 8649 | 67.3 | 4735 | 50.1 | 5024 | 57.7 | 5174 |
| AutoThink | 83.5 | 2017 | 29.7 | 7084 | 70.2 | 3499 | 51.2 | 4606 | 58.6 | 3825 |
| AdaptThink | 81.6 | 1580 | 23.9 | 6432 | 63.2 | 2860 | 48.5 | 4616 | 54.3 | 3871 |
| Base | 83.1 | 4031 | 30.3 | 12150 | 68.3 | 7222 | 50.0 | 8942 | 57.9 | 8086 |
| + Acc-Only | 83.3 | 3061 | **31.6** | 10628 | 70.5 | 5307 | 50.6 | 6402 | 59.0 | 6349 |
| + LAPO-D | **84.7** | 2566 | 28.5 | 8415 | **72.2** | 4132 | 51.3 | 5595 | **59.2** | 5177 |
| + LAPO-I | 84.3 | 2354 | 29.3 | 8318 | 71.2 | 3568 | **51.7** | 4863 | 59.1 | 4775 |
| *Base model: DeepScaleR-1.5B-Preview* | | | | | | | | | | |
| L1-Exact | 80.6 | 1953 | 24.4 | 2625 | 70.9 | 2177 | 48.8 | 2357 | 56.2 | 2278 |
| L1-Max | 81.9 | 1673 | 24.9 | 3638 | 72.7 | 2705 | 50.5 | 2151 | 57.5 | 2541 |
| ThinkPrune-I2k | 85.5 | 1707 | 34.9 | 5095 | 74.3 | 2913 | 54.7 | 3498 | 62.3 | 3303 |
| ThinkPrune-4k | **86.6** | 2042 | 35.5 | 6488 | 76.3 | 3839 | 55.7 | 4010 | 63.5 | 4094 |
| HAPO | 84.4 | 2370 | 31.4 | 7702 | 70.3 | 4301 | 51.4 | 4571 | 59.3 | 4736 |
| AutoThink | 84.9 | 1635 | 36.2 | 7201 | 67.8 | 3658 | 52.5 | 4085 | 60.4 | 4144 |
| Thinkless | 81.3 | 2944 | 28.9 | 9143 | 65.7 | 5276 | 50.2 | 6057 | 56.5 | 5855 |
| Base | 85.8 | 3280 | 35.5 | 9246 | 74.2 | 6416 | 54.6 | 5974 | 62.5 | 6229 |
| + Acc-Only | 85.6 | 2510 | 36.9 | 7319 | 77.6 | 4244 | 55.6 | 4712 | 63.9 | 4696 |
| + LAPO-D | 86.4 | 2365 | 37.6 | 5945 | 77.6 | 3655 | 56.1 | 4499 | 64.4 | 4116 |
| + LAPO-I | 86.3 | 2168 | **38.1** | 5371 | **78.3** | 3765 | **56.3** | 4024 | **64.8** | 3832 |

Table 2: Results with different length guidance for LAPO-I.

| **Method** | MATH-500 | | AIME2024 | | AMC-23 | | OlympiadBench | | Average | |
|---|---|---|---|---|---|---|---|---|---|---|
| | Pass@1 | #Tok | Pass@1 | #Tok | Pass@1 | #Tok | Pass@1 | #Tok | Pass@1 | #Tok |
| *Base model: DeepScaleR-1.5B-Preview* | | | | | | | | | | |
| Base | 85.8 | 3280 | 35.5 | 9246 | 74.2 | 6416 | 54.6 | 5974 | 62.5 | 6229 |
| LAPO-D | 86.4 | 2365 | 37.6 | 5945 | 77.6 | 3655 | 56.1 | 4499 | 64.4 | 4116 |
| w/ Exact | 86.3 | 2168 | **38.1** | 5371 | **78.3** | 3765 | **56.3** | 4024 | **64.8** | 3832 |
| w/ Range | 86.6 | 2153 | 36.5 | 6095 | 76.9 | 3600 | 56.2 | 4011 | 64.1 | 3964 |
| w/ Outside | 86.5 | 2251 | 36.4 | 5882 | 76.3 | 3850 | 55.4 | 4105 | 63.9 | 4022 |
| w/ Implicit | **86.9** | 2181 | 36.2 | 5963 | 76.1 | 4002 | 55.1 | 4206 | 63.6 | 4088 |

(whether it's part of the model's internal thought process). We compare our default approach (w/ Exact) against three variants: w/ Range (less precise guidance), w/ Outside (placing the guidance before `<think>`), and w/ Implicit (no guidance, relying only on the reward). As shown in Table 2, the results demonstrate that both form and position are critical for effective internalization.

Our default method outperforms the less precise Range variant, indicating that specific targets discovered in Discovery stage provide a stronger learning signal. More critically, the guidance's position determines whether the model internalizes a plan or merely follows instructions. Moving the guidance outside the `<think>` block transforms it into an external command and causes accuracy to drop significantly to 63.9%. This illustrates that the model performs best when the budget is framed as part of its own cognitive plan. Finally, removing the guidance entirely results in the worst performance, with accuracy dropping to 63.6% and token count reverting to the LAPO-D baseline. This indicates that our explicit, properly-positioned, self-declarative guidance is the critical mechanism for internalization.

Table 3: Results within different statistical metrics used for target length selection in LAPO-I.

| Method | MATH-500 | | AIME2024 | | AMC-23 | | OlympiadBench | | Average | |
|---|---|---|---|---|---|---|---|---|---|---|
| | Pass@1 | #Tok | Pass@1 | #Tok | Pass@1 | #Tok | Pass@1 | #Tok | Pass@1 | #Tok |
| *Base model: DeepScaleR-1.5B-Preview* | | | | | | | | | | |
| Base | 85.8 | 3280 | 35.5 | 9246 | 74.2 | 6416 | 54.6 | 5974 | 62.5 | 6229 |
| LAPO-D | **86.4** | 2365 | 37.6 | 5945 | 77.6 | 3655 | 56.1 | 4499 | 64.4 | 4116 |
| w/ Median | 86.3 | 2168 | **38.1** | 5371 | **78.3** | 3765 | 56.3 | 4024 | **64.8** | 3832 |
| w/ Mean | 85.6 | 2308 | 36.8 | 6030 | 77.4 | 3658 | **56.6** | 4164 | 64.1 | 4040 |
| w/ Minimum | 85.9 | 2031 | 36.3 | 6080 | 76.7 | 3324 | 55.0 | 3851 | 63.5 | 3821 |

## 5.3 ABLATION ON STATISTICAL METRICS FOR TARGET LENGTH

The choice of a statistical measure to derive the target length $n$ from the distribution of successful solutions is critical. We conduct an ablation study comparing three strategies for this selection (Table 3): using the median (our default), the mean, and the minimum length.

The median proves most effective, achieving the highest average accuracy (64.8%). The mean, being sensitive to long-tail outliers, sets overly generous budgets and slightly reduces accuracy. Conversely, targeting the minimum length, while most token-efficient, causes a significant accuracy drop to 63.5%. This finding validates our hypothesis that pursuing the shortest solution can lead to harmful "over-shortening" and underscores the median's robustness in identifying a typically effective reasoning depth.

## 5.4 QUALITATIVE REFINEMENT OF REASONING BEHAVIORS

This shift towards efficiency is also reflected in the model's qualitative reasoning patterns. We analyzed the frequency of keywords indicative of different cognitive behaviors (Figure 3), revealing a significant shift in the model's reasoning style. The most notable change is a dramatic reduction in keywords associated with "Self-Correction" and "Exploration". LAPO training significantly curtails this verbose, deliberative internal monologue, effectively discouraging redundant verification and exploration. Crucially, keywords for "Context Setting" and "Conclusion Drawing" remain stable. This shows LAPO selectively prunes inefficient, hesitant thought patterns while preserving the essential scaffolding of a logical argument, a behavior further refined in the internalization stage.

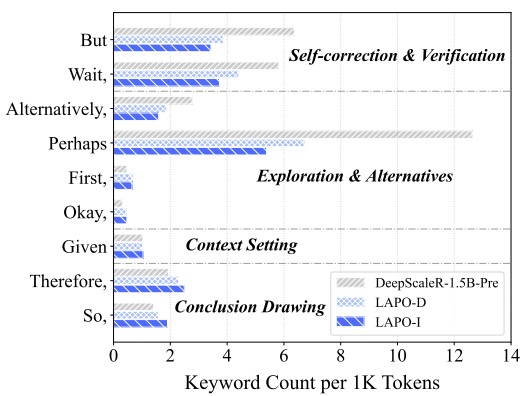

Figure 3: Keyword usage of reasoning behaviors across different stages.

## 6 CONCLUSION

In this work, we introduce Length-Adaptive Policy Optimization (LAPO), a two-stage reinforcement learning framework that enables language models to adjust reasoning length based on problem complexity. Unlike existing approaches that impose uniform constraints, LAPO recognizes that efficient reasoning requires understanding problem-specific computational needs rather than following rigid rules. Our two-stage design enables a natural progression: models first learn what constitutes appropriate reasoning depth through experience, then develop the ability to anticipate these requirements proactively. This approach mirrors how human experts develop intuition about problem complexity, allocating mental effort proportionally to task demands. Extensive experiments validate LAPO's effectiveness. When models learn from their own successful patterns rather than arbitrary constraints, they develop more robust and efficient reasoning strategies.

## REPRODUCIBILITY STATEMEMT

We have made every effort to ensure the reproducibility of our work by providing detailed descriptions of our methodology, experimental setup, and resources. Our proposed Length-Adaptive Policy Optimization (LAPO) framework is described in detail in Section 3, with a complete pseudocode provided in Algorithm 1. All experiments were implemented using the publicly available OpenRLHF framework, and we will release our full source code, configuration files, and training scripts upon publication to facilitate direct replication. The composition of our training dataset and the selection of evaluation benchmarks are outlined in Section 4, with a further analysis of data choices in Appendix A.4. No new data were generated during this study. All analyzed datasets are publicly available and cited appropriately. A comprehensive list of all hyperparameters for training and evaluation can be found in Table 4 of Appendix A.2. This section also contains the exact prompt templates used for both stages of LAPO and for all ablation studies. Furthermore, to aid in validating the training process, we present an analysis of the training dynamics in Appendix A.3, showcasing the learning curves for key metrics.

## ETHICS STATEMENT

This work does not involve human subjects, personal data, or any other form of sensitive information. All datasets used in our experiments—including the DeepScaleR-Preview-Dataset, MATH, AIME2024, AMC23, Olympiad-Bench, and GPQA—are publicly available benchmark datasets designed for evaluating reasoning capabilities in large language models. We have strictly adhered to ethical research practices, and our work relies exclusively on pre-existing, public data, thereby raising no concerns regarding privacy, security, or fairness from data collection.

Our method, Length-Adaptive Policy Optimization (LAPO), focuses on improving the computational efficiency and reasoning accuracy of language models. It is a foundational technique that does not inherently introduce new risks of harmful applications. To the best of our knowledge, this research complies with the ICLR Code of Ethics and poses no foreseeable ethical concerns.

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

# A APPENDIX

## A.1 LLM USAGE

We wish to clarify that a large language model (LLM) provided assistance in the preparation of this manuscript. Its use was confined exclusively to enhancing the language, including grammar, style, and overall readability. The LLM made no substantive contributions to the research design, analysis, or the formulation of our conclusions.

## A.2 IMPLEMENTATION DETAILS

**System prompt used for training.** The system prompts used for the two-stage training are shown in the boxes below. The prompt titled **LAPO-D-prompt** was used for DeepSeek-R1-Distill-Qwen-1.5B, and **LAPO-I-prompt** was used for DeepScaleR. This approach maintains consistency with the original RL training of DeepSeek-R1.

---

**LAPO-D-prompt**

You are a helpful assistant. A conversation between User and Assistant. The user asks a question, and the Assistant solves it. The Assistant first thinks about the reasoning process in the mind and then provides the user with the answer. The reasoning process is enclosed within <think> and </think> tags, respectively, i.e., <think> reasoning process here </think> answer here. User: {question} Please think step by step and output the final answer within \boxed{}. Assistant: <think>

---

**LAPO-I-prompt**

You are a helpful assistant. A conversation between User and Assistant. The user asks a question, and the Assistant solves it. The Assistant first thinks about the reasoning process in the mind and then provides the user with the answer. The reasoning process is enclosed within <think> and </think> tags, respectively, i.e., <think> reasoning process here </think> answer here. User: {question} Please think step by step and output the final answer within \boxed{}. Assistant: <think> I will answer the question with {length} tokens.

---

**Prompts for Ablation Studies.** To support the ablation studies in Section 5.2, we utilized several variations of the prompt structure. These are detailed below:

**w/ Range:** For this variant, the self-declarative statement was modified to specify a range instead of an exact number. The prompt concluded with:

"... Assistant: <think> I will answer the question within approximately $n_{min}$ to $n_{max}$ tokens."

The length reward for this configuration was a uniform score of 1.0 for any correct solution whose length fell within this range.

**w/ Outside:** In this configuration, the length guidance was provided as an external instruction before the <think> token, altering the prompt to:

"... User: {question} Please think step by step, answer the question with $n$ tokens, and output the final answer within boxed{}. Assistant: <think>"

**w/ Implicit:** This variant used the same prompt as the Discovery stage (LAPO-D-prompt), with no explicit length guidance provided to the model. However, the reward function was the same as the Internalization stage (Eq. 6), based on the target length $\mathcal{M}(q)$.

**Training and Reproduction Details.** We trained the model on the OpenRLHF framework. During training, we sampled 8 responses for each query in the batch with a temperature of 1.0, set the kl parameter to 0.0001, used a learning rate of 1e-6 and a batch size of 128, and set the maximum context length to 4K tokens during training. Both LAPO-D and LAPO-I training were conducted for 3 episodes, approximately 240 steps. The $\alpha$ and $\beta$ parameters in $R_1$ and $R_2$ were 0.7 and 0.7, respectively. All experiments were conducted using 4 A800 GPUs. We provide training hyperparameters in Table 4.

**Discussion on Hyperparameter Selection.** Our methodology incorporates several hyperparameters to guide the learning process. Our choices are based on principled heuristics designed to ensure stable and effective training. The percentile range for the length target, [P30, P70], was selected to define a robust zone of reasonableness, filtering out statistically anomalous short solutions that may be correct by chance, while also discouraging excessive verbosity. This approach is intentionally more stable than targeting only the minimum length. The reward weights, $\alpha$=0.7 and $\beta$=0.7, were set to provide a substantial learning signal for efficiency, yet remain subsidiary to the primary binary reward for correctness, thereby mitigating the risk of reward hacking. For penalizing deviations outside the target range, a linear decay function, f(d), was employed to be less aggressive than exponential alternatives, allowing for necessary flexibility on complex problems. In the Internalization stage, the Gaussian standard deviation $\sigma$=120 creates a soft adherence target, tolerating minor deviations from the self-proposed plan while penalizing large ones, thus balancing planning with execution flexibility. Finally, the number of episodes for each stage (E1=3, E2=3) was determined empirically, as we observed that model performance on both efficiency and accuracy metrics stabilized after this duration, as illustrated by our training dynamics in Figure 4a and 4b.

Table 4: Training Hyperparameters

| Hyperparameter | Value |
| --- | --- |
| Epochs | 1 |
| Episodes | 3 |
| Learning Rate | 1e-6 |
| Train Batch Size | 128 |
| Temperature | 1.0 |
| Rollout per Prompt | 8 |
| Prompt Max Length | 1024 |
| Generation Max Length | 4096 |
| KL Coefficient | 0.0001 |
| Precision | BF16 |
| $\alpha$ | 0.7 |
| $\beta$ | 0.7 |
| $\sigma$ | 120 |

### A.3 TRAINING DYNAMICS

We analyze the training dynamics by periodically evaluating model checkpoints on the MATH-500 validation set to understand the learning mechanisms of our two-stage framework. As illustrated in Figures 4a and 4b, LAPO achieves a superior balance between efficiency and accuracy across both training stages.

**Continuous Efficiency Gains.** Figure 4a shows a clear, two-step reduction in token generation. In Stage 1, the LAPO-D policy rapidly becomes more concise, with its average length decreasing from a verbose baseline of 3,280 tokens to a stable 2,365 tokens, driven by the length-aware reward ($R_1$). Building on this, the LAPO-I policy achieves further compression, reducing the length to below 2,200 tokens. This demonstrates that the plan-adherence reward ($R_2$), combined with in-context guidance, effectively encourages the model to execute its self-proposed reasoning plans more precisely.

**Accuracy Maintenance and Refinement.** Crucially, these efficiency gains do not compromise performance. As shown in Figure 4b, accuracy on MATH-500 is consistently maintained or improved. The LAPO-D policy's accuracy climbs from 85.8% to over 86.4%, suggesting the reward mechanism prunes redundant or error-prone reasoning steps. The LAPO-I policy sustains this high accuracy level even on a much tighter token budget. Notably, it exhibits a transient performance peak, a key finding that suggests the in-context guidance actively steers the model toward more focused and effective reasoning, rather than merely acting as a constraint.

In summary, the training dynamics validate our two-stage design. LAPO-D establishes a robust foundation for efficient reasoning, which LAPO-I then refines to achieve a superior performance-cost balance. The smooth convergence on a challenging validation set confirms that by learning from its own successful patterns, the model develops transferable and efficient reasoning strategies.

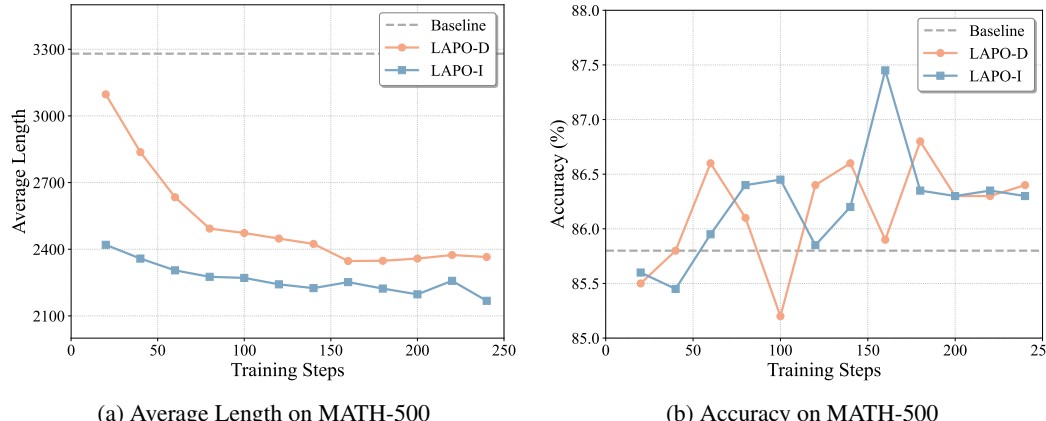

(a) Average Length on MATH-500        (b) Accuracy on MATH-500

Figure 4: Training dynamics evaluated on the MATH-500 validation set. Checkpoints were saved periodically during training on our mixed dataset. (a) Both LAPO-D and LAPO-I policies learn to significantly reduce the average response length. (b) These efficiency gains are achieved while maintaining or even improving accuracy over the baseline.

Table 5: Ablation study on the training dataset. This table compares performance when trained on different data sources. For each metric column, **bold** indicates the best score and underline indicates the second-best score across all configurations.

| Method | MATH500 | | AIME2024 | | AMC-23 | | OlympiadBench | | Average | |
|---|---|---|---|---|---|---|---|---|---|---|
| | Pass@1 | #Tok | Pass@1 | #Tok | Pass@1 | #Tok | Pass@1 | #Tok | Pass@1 | #Tok |
| *Training Data: Combined (Ours)* | | | | | | | | | | |
| LAPO-D | 86.4 | 2365 | 37.6 | 5945 | 77.6 | 3655 | 56.1 | 4499 | 64.4 | 4116 |
| LAPO-I | 86.3 | 2168 | **38.1** | 5371 | **78.3** | 3765 | **56.3** | 4024 | **64.8** | 3832 |
| *Training Data: DeepScaleR-only* | | | | | | | | | | |
| LAPO-D | 86.1 | 2397 | 36.8 | 6153 | 76.8 | 3983 | 55.5 | 4258 | 63.8 | 4197 |
| LAPO-I | 86.1 | 2210 | 36.5 | 6418 | 77.0 | 3791 | 55.6 | 3933 | 63.8 | 4088 |
| *Training Data: MATH-only* | | | | | | | | | | |
| LAPO-D | **86.5** | 2398 | 38.0 | 7034 | 77.3 | 4060 | 55.8 | 4494 | 64.4 | 4496 |
| LAPO-I | 86.1 | 2340 | 35.5 | 6452 | 75.8 | 4021 | 54.5 | 4194 | 63.0 | 4251 |

## A.4 SELECTION OF TRAINING DATASET

As mentioned in section 4 Experiment Setup, we chose a mixed dataset for training in our experiments. In this section, we provide a detailed analysis of the impact of different dataset selections on model performance. Table 5 shows the test results on various benchmarks after two-stage training using different training datasets. Several important findings can be observed from the experimental results. Combined-data achieved the best performance in terms of average accuracy, showing a clear advantage over single-dataset training. This indicates that a dataset with a balanced difficulty distribution helps enhance the model's generalization ability across different types of questions. In terms of token usage efficiency, the model trained on combined-data also performed the best. This suggests that problems with different difficulty gradients help establish a more accurate complexity-length mapping relationship. By exposing the model to a wider range of problem difficulties, it can better learn the optimal thinking range for different questions. Taking all these factors into consideration, we selected the mixed dataset as the training data to expose the model to a more diverse set of problems and enable it to deeply learn the optimal reasoning patterns for different questions.

Table 6: Performance on the GPQA benchmark. LAPO demonstrates generalizable efficiency and accuracy gains in a non-mathematical, knowledge-intensive domain.

| Method | Pass@1 (%) | #Tokens |
|---|---|---|
| *Base Model: DeepSeek-R1-1.5B* | | |
| Base | 36.1 | 10297 |
| + LAPO-D | **38.1** | 7596 |
| + LAPO-I | 36.9 | 7235 |
| *Base Model: DeepScaleR-1.5B-Preview* | | |
| Base | 36.1 | 7667 |
| + LAPO-D | **38.3** | 6176 |
| + LAPO-I | 37.8 | 6154 |

Table 7: Robustness of LAPO-I to conflicting length instructions on MATH-500.

| Method | Length Constraint | MATH-500 | |
|---|---|---|---|
| | | Pass@1 (%) | #Tok |
| *LAPO-I* | | | |
| Base | N/A | **86.3** | 2168 |
| +Short | 500 | 86.0 | 2279 |
| +Long | 3500 | 85.9 | 2300 |
| *LAPO-I w/ Outside* | | | |
| Base | N/A | **86.2** | 2251 |
| +Short | 500 | 85.1 | 1247 |
| +Long | 3500 | 86.1 | 2821 |

## A.5 GENERALIZABILITY TO EXPERT-LEVEL QUESTION ANSWERING.

To test if LAPO's benefits extend beyond structured mathematical reasoning, we evaluated our method on the GPQA benchmark. The results, presented in Table 6, demonstrate that LAPO's core principles are highly generalizable.

For both base models, LAPO achieves a compelling dual improvement in accuracy and efficiency. On the DeepSeek-R1-1.5B model, LAPO-D improves Pass@1 accuracy by a significant 2.0 points while reducing token generation by 26.2%. Similarly, on the more advanced DeepScaleR-1.5B-Preview, LAPO-D boosts accuracy by 2.2 points and cuts tokens by 19.4%. The internalization stage consistently pushes efficiency further while maintaining a strong accuracy improvement over the baseline. This robust performance on a knowledge-intensive, non-mathematical task indicates that LAPO is not merely exploiting domain-specific patterns. Instead, it learns a fundamental and transferable skill: how to allocate cognitive effort efficiently for complex reasoning across different domains.

## A.6 ANALYSIS OF INTERNALIZATION

To validate that LAPO fosters genuine internalization, we stress-tested our default LAPO-I model against the w/ Outside ablation variant using adversarial Short (500 tokens) and Long (3500 tokens) length prompts. The results in Table 7 reveal a stark behavioral divergence. Our default LAPO-I remains robust, its output length staying stable around its 2200-token baseline, thus ignoring the conflicting external instructions. In contrast, the w/ Outside model is clearly influenced: its token count drops to 1247 under the Short constraint and rises to 2821 under the Long one. This comparison indicates that the placement of guidance is critical. Framing the budget as part of the model's internal plan (inside `<think>`) builds a robust, internalized behavior. Framing it externally teaches superficial instruction-following. This indicates the observed robustness of LAPO-I is a direct result of our internalization mechanism.

## A.7 DIFFICULTY-AWARE COMPUTATIONAL ALLOCATION

To understand the mechanisms behind LAPO's efficiency gains, we examine its ability to allocate computational resources in proportion to problem complexity. We evaluate LAPO-trained models on benchmarks with clear difficulty gradients, from MATH Level 1 up to the highly complex AIME 2024. As shown in Figure 5, our models demonstrate a remarkable emergent capability for difficulty-aware resource allocation. There is a clear, near-linear positive correlation between problem complexity and the average reasoning length. On simpler problems, the models generate concise responses, while for the most challenging AIME questions, they produce extensive reasoning chains that are substantially longer than any solution observed during the training phase. This ability to extrapolate reasoning depth well beyond the bounds of their training experience is a crucial finding. It provides strong evidence that LAPO does not merely teach models to compress their outputs.

Instead, it successfully imparts a generalizable principle of complexity-to-length mapping. This allows the models to dynamically and appropriately scale their computational investment when faced with novel problems of varying difficulty. The consistent scaling behavior across different base models further underscores that LAPO develops a robust, fundamental reasoning strategy rather than model-specific optimizations.

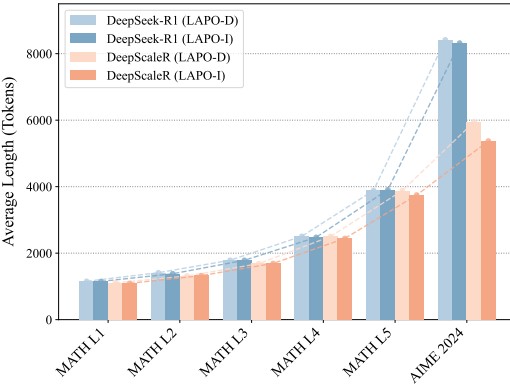

Figure 5: Reasoning length allocation across mathematical problem difficulty levels. LAPO learns to scale computation with complexity.

