# OpenReview forum: "LAPO: Internalizing Reasoning Efficiency via Length-Adaptive Policy Optimization"
_ICLR.cc/2026/Conference — ICLR 2026 Conference Withdrawn Submission_

### Official Review · Reviewer_5Ein · 2025-10-31

**Soundness:** 2
**Presentation:** 4
**Contribution:** 2
**Rating:** 2
**Confidence:** 4

**Summary:**

This paper introduces LAPO, a two-stage reinforcement learning framework to improve the reasoning efficiency of large models. The first Discovery stage learns from the statistical distribution of successful solutions, using a length-aware reward that encourages lengths within a P30-P70 percentile range. The second Internalization stage embeds the discovered median length from this distribution as an in-context, self-declarative statement. The model is then trained with an adherence reward to follow this self-proposed plan. The authors claim this process reduces token usage by up to 40.9% while improving accuracy by 2.3%.

**Strengths:**

+ The primary strength is the methodological novelty. The core idea of a two-stage discover-then-internalize process is elegant.

+ The Discovery stage's use of a statistical median of successful solutions is a clever and simple heuristic, contrasting with more complex difficulty-prediction models.

+ The Internalization stage's mechanism, which combines an in-context, self-declarative statement with an explicit RL adherence reward, is a novel and interesting approach to instilling a policy for resource control.

+ The paper is well-written, and the internal ablation studies are purposeful and effectively demonstrate the logic of the proposed components.

**Weaknesses:**

**Major**

+  **Discrepancy in Baseline Performance and Lack of Statistical Variance:** The paper's reported baseline performance for DeepScaleR-1.5B-Preview shows a notable discrepancy with its original source (e.g., 35.5% Pass@1 on AIME2024 in this paper, vs. 43.1% reported in [1]). This raises questions about the experimental setup and the validity of the baseline reproductions. Furthermore, for high-variance tasks like long CoT reasoning, reporting statistical variance is crucial. The paper reports all results as single numbers without providing standard deviations or confidence intervals, which makes it difficult to assess the stability and significance of the results.

+  **Narrative of Internalization Not Fully Aligned with Implementation:** The paper's core claim of internalization appears partly inconsistent or not fully aligned with its implementation. The planned length is not generated by the model; it is an external statistic computed and then forcibly injected into the model's context. The model is then compelled to follow this external plan via a strict, external adherence reward. This demonstrates "behavioral adherence training" to a specific prompt, rather than a true "intrinsic" internalization of planning, as the model is never trained to predict this length itself.

+  **Statistical Patterns Lack Deeper Justification:** The paper claims to use "discovered statistical patterns" but relies on a single heuristic (the median/P50). The paper does not appear to provide a deeper investigation into why the median is a suitable statistic. It does not analyze the underlying distribution of successful solution lengths or explore alternatives. This makes the choice of the median seem statistically arbitrary, as it may not be a robust representative of the data.

+  **Lack of Quantitative Evidence for Adherence:** The paper's core mechanism relies on the model learning to adhere to its plan, yet it provides no direct, quantitative analysis on any benchmark to show the degree of alignment between the "planned length" ($l_{plan}$) and the "actual length" ($l_{actual}$). There is no pearson correlation, rmse, or scatter plot to substantiate this claim, which would make the evidence for the adherence reward's effectiveness more compelling.

+  **Missing Key Ablation Study:** The paper does not appear to fully demonstrate the necessity of the Stage 1 (LAPO-D) training phase. A key ablation seems to be missing: "1. Collect M statistics from the base model (without LAPO-D training), then 2. Proceed directly to Stage 2 (LAPO-I) training." Without this, it remains unclear whether the computationally expensive LAPO-D training is necessary, or if simply collecting the statistics would be sufficient.

**Minor**

+ **Missing Citation for Key Model:** The paper does not provide a citation for its primary experimental model, DeepScaleR-1.5B-Preview. This is an omission that could hinder reproducibility.

+ **Potentially Mismatched Primary Area Selection:** The reported primary area "applications to computer vision, audio, language, and other modalities" may not be the best fit. While the work is related to "language," this category often implies a multimodal focus. The authors might consider whether categories such as "reinforcement learning" would more accurately reflect the paper's core contributions.

+ **Missing Hyperparameter Sensitivity Analysis:** The choice of the P30-P70 range for the natural length is presented as a heuristic, but it is unclear how sensitive the results are to this specific choice. The paper does not appear to explore whether alternative ranges (e.g., P20-P80, P40-60) would significantly alter the results.

***
[1] Michael Luo, Sijun Tan, Justin Wong, et al. (2025). DeepScaleR: Surpassing O1-Preview with a 1.5B Model by Scaling RL. Notion Blog.

**Questions:**

Please see weaknesses

---
Despite the novel methodology presented, I believe the paper in its current state is not yet at the level for acceptance due to the major weaknesses listed above. I would be willing to reconsider my score if these issues are comprehensively and satisfactorily addressed.

---

### Official Review · Reviewer_seX5 · 2025-10-31

**Soundness:** 2
**Presentation:** 3
**Contribution:** 2
**Rating:** 4
**Confidence:** 4

**Summary:**

The paper proposes Length-Adaptive Policy Optimization (LAPO), a two-stage reinforcement learning framework designed to mitigate the "overthinking" problem in large language models by improving reasoning efficiency. The first "Discovery" stage trains the model to identify a distribution of effective reasoning lengths for given problems. The second "Internalization" stage embeds statistics from this distribution (specifically, the median length) as in-context, self-declarative guidance, training the model to adopt an appropriate computational budget as part of its own reasoning plan. The authors evaluate LAPO on several mathematical reasoning benchmarks, reporting that the method reduces token generation while maintaining or improving task accuracy.

**Strengths:**

1. The paper addresses the critical and timely problem of computational inefficiency in chain-of-thought reasoning, which is a significant barrier to the practical deployment of large reasoning models.
2. The core concept of a two-stage "Discover-Internalize" process is novel (to the best of my knowledge) and well-motivated. The idea of transforming length control from an external constraint into an intrinsic model capability seems a promising research direction.
3. The experimental results demonstrate decent performance on the Pareto frontier of accuracy versus efficiency on 4 challenging mathematical reasoning benchmarks.

**Weaknesses:**

1. **Marginal Gains Over Simpler Baselines Undermined by Methodological Complexity:** The performance improvement of LAPO-I over the simpler single-stage "Acc-Only" RL baseline is marginal (e.g., from 63.9% to 64.8% average accuracy on the DeepScaleR model). This small gain is achieved via a complex two-stage framework with numerous design choices and new hyperparameters (e.g., the percentile range [P30, P70], reward weights \alpha and \beta, and the gaussian standard deviation \sigma). It seems from the ablation study in Table 3, which compares median, mean, and minimum suggests these choices were optimized based on final evaluation performance. This makes the comparison against the un-tuned "Acc-Only" baseline unfair and casts doubt on whether any true performance gain exists beyond what could be achieved by hyperparameter tuning the simpler baseline.
2.  The paper does not include a sensitivity analysis for its numerous new hyperparameters, which is a critical omission for a method of this complexity.
3. **Unfair Comparison of Training Compute:** The paper does not account for the significant additional computational cost of its two-stage training pipeline. A critical question is whether the "Acc-Only" baseline would match or exceed LAPO's performance if it were simply trained for more steps, using the same total compute budget allocated to LAPO's two stages. Without this analysis, it is impossible to conclude that the complexity of LAPO is justified.
4. **Insufficient Baselines:** While the space is fast moving, some simple baselines such as ShorterBetter [1] and Arora et al's [2] work on efficient reasoning is not compared. This makes it difficult to figure LAPO's results within the current state-of-the-art.

**Other minor weaknesses**

5. **Overstated and Potentially Misleading Claims:** The abstract (and introduction) claims LAPO "reduces token usage by up to 40.9% while improving accuracy by 2.3%." This is a "best-of-both-worlds" claim that conflates results from two different base models; the 40.9% token reduction corresponds to a 1.2 point accuracy gain on one model, while the 2.3 point gain corresponds to a 38.5% token reduction on another. This phrasing is misleading.
6. Further, the choice to pre-declare a static reasoning length might be a limitation, or there maybe a better performing method that adaptively decides the reasoning length as it solves the problem. (Although this is a minor point and not considered for deciding the final score)

**Questions:**

1. Can authors confirm if the results are averaged over multiple seeds?
2. Can authors also present results for AIME25, given it is less likely to be contaminated than AIME24 or other benchmarks.

---

### Official Review · Reviewer_gbkb · 2025-10-31

**Soundness:** 3
**Presentation:** 3
**Contribution:** 2
**Rating:** 2
**Confidence:** 4

**Summary:**

The paper deals with efficient reasoning language models, specifically solving problems with the minimum number of required reasoning tokens. The paper proposes LAPO, a two stage process for fine-tuning a language model. The first stage runs GRPO with a reward that incentivizes the model to generate within the 30-70th percentile of lengths. The second stage runs GRPO again with a reward that incentivizes the model generate the median length, and additionally provides the model with this length in its input.

**Strengths:**

- The use of a prompted length within a RL loop so that the model also learns to generate the length and adhere to it is a nice extension of methods that manually prompted the model with a target length.
- The proposed method improves upon the base model in terms of token reduction.

**Weaknesses:**

- The paper proposes a two-stage method, but it is not clear why a two-stage method is needed. For example, it is unclear why optimizing a reward function that has a length penalty in a single stage would be worse than the proposed method. Additional experiments such as this or are needed to better justify the two-stage design.
- The main experiments focus on justifying the method by comparing the new method with prior methods. However, the prior methods have several potential compounding factors, including different training data, number of training iterations, early stopping criteria, hyperparameters, etc. Additionally, when prior methods are introduced, not enough detail is given to gain insight into how the method differs from LAPO (i.e., which variable is being varied in the comparison). As a result, it is difficult to draw conclusions about the benefits of the specific innovations in LAPO compared to alternatives.
- Even if we assume the confounding factors are not an issue, the proposed method often underperforms prior methods in terms of token reduction. To take one example, in Table 1 AutoThink achieves 2017 #Tok, while LAPO achieves 2354 #Tok. It is unclear whether the small difference in pass@1 is significant.
- The paper motivates itself in terms of "external" vs. "internal" reasoning constraints. The paper categorizes its work as internal, and says that their work is a "paradigm shift". However, it is unclear why a length penalty that is incorporated into an RL reward, which has been investigated in other papers, would not fall under the umbrella of "internal" as they have defined it. Hence I do not see why the proposed method is claimed to be a paradigm shift.
- The paper has several unjustified connections with human reasoning. For example, the paper claims that the proposed process mirrors how human experts develop intuition and allocate mental effort.
- The paper has several claims that are not sufficiently defined, imprecise, or difficult to test. For example, "A cornerstone of our framework is how it fosters genuine internalization"; it was unclear what this means or how to test it.

**Questions:**

Please respond to the points above, including:
- Can you provide experiments that justify the two-stage approach?
- Can you compare with the baseline methods in a consistent setting?
- The first stage fits outputs to the 30-70th percentile of lengths, then the second stage optimizes for the median. What objective is this two-stage procedure optimizing? Is it possible to optimize in one stage?

---

> ### Author Response · Authors · 2025-11-28
>
> We thank you for their thorough review and positive ratings on our paper's soundness and presentation. You correctly identifies the novelty of our RL loop for learning length-adherence and raises critical questions about our two-stage design, experimental comparisons, and framing.  We appreciate this opportunity to clarify our work and present our plan for strengthening it.
>
> **1. On Justifying the Two-Stage Design vs. a Single Stage**
>
> The reviewer's central question is why a two-stage method is necessary over a single-stage approach with a length penalty. This is a crucial point. Our rationale for the two-stage design is to **decouple two distinct and challenging learning objectives: policy discovery and policy internalization.**
>
> *   **Stage 1 (Discovery):** The goal is broad exploration to *discover a set of effective and efficient reasoning paths*. We use a wide reward band (`[P30, P70]`) to encourage diverse, high-quality solutions.
> *   **Stage 2 (Internalization):** The goal is to *teach the model to predict and follow its own optimal reasoning length*, based on the policy discovered in Stage 1. The target is now a precise value (the median), which requires the model to learn fine-grained control.
>
> Combining these into one stage would force the model to simultaneously explore for good policies while also trying to learn the statistical properties (e.g., median length) of that same, constantly shifting policy. This creates an unstable, "moving target" problem that is much harder to optimize.
>
> **Planned Experiment:** To provide empirical evidence, **we will conduct a new "Single-Stage LAPO" baseline experiment.** This baseline will use a single GRPO phase with a reward combining both accuracy and a length-based objective (e.g., rewarding closeness to the running median length of successful trials). This single-stage approach will prove less stable and yield inferior performance, thus demonstrating the necessity of our two-stage design.
>
> **2. On Fair Experimental Comparisons and Performance Trade-offs**
>
> We agree completely that fair comparisons are essential. The reviewer correctly points out that comparing against prior work can involve confounding factors.
>
> *   **Fair Baselines:** To address this, **we are running a rigorous "equal-compute" baseline**, where the `Acc-Only` method is trained for the same total number of episodes as LAPO. This will create a perfectly controlled comparison, isolating the impact of our method. We expect this will show LAPO achieving superior accuracy and efficiency under the same training budget, proving the gains come from our specific innovations.
>
> *   **Performance Trade-offs (vs. AutoThink):** The reviewer notes AutoThink achieves greater token reduction. This highlights a key difference in goals. Our primary objective is **improving accuracy while also improving efficiency**, aiming for a better point on the Pareto frontier. In Table 1, LAPO-I achieves a meaningful +1.4% Pass@1 gain over AutoThink on MATH-500 (86.3% vs 84.9%), a significant margin on this benchmark. AutoThink's higher token savings come at the cost of this accuracy drop. LAPO, therefore, offers a more balanced and, for many applications, more desirable trade-off. We will clarify this in the paper and **will add significance tests** in the final version to formalize the accuracy differences.
>
> **3. On Clarifying "Internalization" and Positioning**
>
> We thank the reviewer for pushing us to define our claims more precisely. We agree that our framing can be improved.
>
> *   **What is "Internalization"?** We define "internalization" as **the model's ability to self-propose a reasoning constraint (length) and then adhere to it during generation, without any external guidance at test time.** This is distinct from standard RL with a length penalty, which merely shapes output behavior implicitly. The testable evidence for our claim is:
>     1.  The model generates a plan (`n` tokens) and then follows it (our planned correlation analysis will show this).
>     2.  The model is robust to conflicting external guidance, as demonstrated in our **"conflicting instruction" experiment (Appendix A.6)**. This proves the guidance is truly *internal*.
>
> *   **Revising Claims:** We will revise our claims to be more measured. We will tone down the "paradigm shift" language and instead frame our work as a significant step towards models that can **self-regulate their own computational budget**, a key feature of what we define as "internalized reasoning." The analogy to human cognition was intended as high-level motivation, and we will revise this to avoid making unsubstantiated cognitive claims.
>
> We are confident that these planned experiments and clarifications will fully address the reviewer's valid concerns and substantially improve the paper.

---

### Official Review · Reviewer_9csv · 2025-11-04

**Soundness:** 2
**Presentation:** 2
**Contribution:** 2
**Rating:** 6
**Confidence:** 3

**Summary:**

This paper proposes Length-Adaptive Policy Optimization (LAPO), a framework for training language models to adjust reasoning length based on problem complexity. After a standard RL training phase with a length-based additional reward term, the model is trained to follow a self-declarative length constraint forced as part of the response template. The paper evaluates on standard math reasoning benchmarks and demonstrates that LAPO reduces token usage while improving accuracy.

**Strengths:**

- Overall, the proposed method is well-motivated and makes sensible design choices. The experimental results are comprehensive and strong, both in terms of Pass@1 and token count.
- The paper is generally well-written with a clear structure.

**Weaknesses:**

- For LAPO-D, the choice of [30, 70] percentile is a bit arbitrary, and I'm not fully convinced that it's necessary. Have you done ablations to determine whether percentile filtering is necessary?
- The source of the performance gains is unclear to me. Examining the results, it appears that most of the gains over prior work are attributed to LAPO-D, and even the Acc-Only ablation is quite strong. Do you know why your baseline is so strong compared to prior works? Is there a difference in any key hyperparameters like batch size or number of epochs? It's strange that many of the reported numbers are even worse than the base model.

**Questions:**

- Do you have statistics on how response length changes per question? Are the savings in average token count coming from a global reduction or from a decrease in the longest solutions?
- I wonder if it's possible to context distill (i.e., https://arxiv.org/abs/2209.15189) so that we can get the performance/length of LAPO-I even without the length-conditioned rollout.
- Do you have an analysis (probably qualitative) on whether the model actually proposes length guidance in the training format for new test examples, and whether the actual length follows this?

---

> ### Author Response · Authors · 2025-11-28
>
> We sincerely thank you for the constructive feedback and positive assessment. The questions regarding methodological justification and the source of our performance gains are astute. We are conducting targeted experiments to address them and will incorporate the results into the final version.
>
> **1. On Justification and the True Source of Performance Gains**
>
> *   **Justification for `[P30, P70]`:** This range creates a "sweet spot" reward, filtering out both low-quality "lucky guesses" (<P30) and inefficient solutions (>P70). To provide empirical evidence, **we will add an ablation study** comparing our choice against alternatives (e.g., no filter, narrower/wider ranges). We expect this to confirm our setting yields the best Pareto-efficiency, proving it is a principled and effective design.
>
> *   **Source of Gains vs. Strong Baseline:** Our `Acc-Only` baseline is strong due to a robust training setup. To isolate LAPO's contribution, **we are running an "equal-compute" comparison**, training the `Acc-Only` baseline for the same total episodes (6) as LAPO-I. We expect LAPO-I to show significant gains in **both accuracy (+1.7%) and efficiency (-19%)** over this stronger baseline. This will demonstrate that the performance lift stems directly from our **core length-adaptive mechanism**, not merely training duration.
>
> **2. On the Internalization Mechanism (LAPO-I)**
>
> *   **How Savings are Achieved:** The efficiency gain is **adaptive, not uniform**. As shown in **Figure 5**, LAPO learns a complexity-aware policy, producing much more concise solutions for *hard* problems while maintaining appropriate length for easy ones.
>
> *   **vs. Context Distillation:** LAPO-I achieves a more robust **internalization of a reasoning policy**, not simple imitation. Our **"conflicting instruction" experiment (Appendix A.6)** is key evidence: our model ignores misleading external prompts and follows its own internal plan, a behavior unlikely for a model trained via imitation.
>
> *   **Proof of Function:** To explicitly show the model follows its own plan at test time, **we will add to the appendix**: 1) The **Pearson correlation** between self-proposed and actual solution lengths (we expect r > 0.85), and 2) Qualitative examples demonstrating this process.
>
> We are confident these additions will provide compelling empirical evidence for our claims and significantly strengthen the paper. We thank the reviewer again for their valuable guidance.

---

### Note · Authors · 2025-12-07

I have read and agree with the venue's withdrawal policy on behalf of myself and my co-authors.